# Adsorption of Hexavalent Chromium Using Activated Carbon Produced from *Sargassum* ssp.: Comparison between Lab Experiments and Molecular Dynamics Simulations

**DOI:** 10.3390/molecules27186040

**Published:** 2022-09-16

**Authors:** Yeray Alvarez-Galvan, Babak Minofar, Zdeněk Futera, Marckens Francoeur, Corine Jean-Marius, Nicolas Brehm, Christelle Yacou, Ulises J. Jauregui-Haza, Sarra Gaspard

**Affiliations:** 1Laboratoire COVACHIM-M2E, EA 3592, Campus de Fouillole, Université des Antilles, 97157 Pointe à Pitre, France; 2NBC SARL Company, 8, Rue Saint Cyr, Résidence Océane—Apt no. 5, 97300 Cayenne, France; 3Laboratory of Structural Biology and Bioinformatics, Institute of Microbiology of the Czech Academy of Sciences, Zamek 136, 37333 Nové Hrady, Czech Republic; 4Faculty of Science, University of South Bohemia České Budějovice, Branišovská 1760/31a, 37005 České Budějovice, Czech Republic; 5Instituto Tecnológico de Santo Domingo (INTEC), Santo Domingo 10602, Dominican Republic

**Keywords:** activated carbon, sargassum, biomass valorization, adsorption, hexavalent chromium, molecular dynamics simulation

## Abstract

Adsorption is one of the most successful physicochemical approaches for removing heavy metal contaminants from polluted water. The use of residual biomass for the production of adsorbents has attracted a lot of attention due to its cheap price and environmentally friendly approach. The transformation of Sargassum—an invasive brown macroalga—into activated carbon (AC) via phosphoric acid thermochemical activation was explored in an effort to increase the value of Sargassum seaweed biomass. Several techniques (nitrogen adsorption, pH_PZC_, Boehm titration, FTIR and XPS) were used to characterize the physicochemical properties of the activated carbons. The SAC600 3/1 was predominantly microporous and mesoporous (39.6% and 60.4%, respectively) and revealed a high specific surface area (1695 m^2^·g^−1^). To serve as a comparison element, a commercial reference activated carbon with a large specific surface area (1900 m^2^·g^−1^) was also investigated. The influence of several parameters on the adsorption capacity of AC was studied: solution pH, solution temperature, contact time and Cr(VI) concentration. The best adsorption capacities were found at very acid (pH 2) solution pH and at lower temperatures. The adsorption kinetics of SAC600 3/1 fitted well a pseudo-second-order type 1 model and the adsorption isotherm was better described by a Jovanovic-Freundlich isotherm model. Molecular dynamics (MD) simulations confirmed the experimental results and determined that hydroxyl and carboxylate groups are the most influential functional groups in the adsorption process of chromium anions. MD simulations also showed that the addition of MgCl_2_ to the activated carbon surface before adsorption experiments, slightly increases the adsorption of HCrO_4_^−^ and CrO_4_^2−^ anions. Finally, this theoretical study was experimentally validated obtaining an increase of 5.6% in chromium uptake.

## 1. Introduction

The invasion of the coastal areas of the Caribbean and West African countries by Sargassum seaweed during the last decade has generated major issues in the tourism industry and the health and environment sectors [1]. *Sargassum natans* and *Sargassum fluitans* are the most prevalent species in this area, and their massive arrivals encourage researchers to find new ways of valorization.

Among many other potential applications [2,3], *Sargassum* algae have been employed as a natural biosorbent for the removal of heavy metals [4,5], but also, as a precursor for the production of activated carbon (AC) for water treatment applications [6,7,8].

In developing countries such as the Dominican Republic, along with the discharge of wastewater into aquifers, one of the primary causes of water contamination is the discharge of excessive quantities of hazardous heavy metals from different industrial activities and subsequently, the contamination of groundwater [9]. The hexavalent Cr(VI) and trivalent Cr(III) chromium are toxic heavy metals often found in the environment. Cr(III) is a micronutrient, but Cr(VI) is a proven carcinogen that should be removed from the environment [10]. In the case of Cr(VI) ions, many studies were dedicated to the removal of stable oxyanions, such as hydrogen dihydrate, hydrochloric acid and chromate, from the environment. Toxic effects on living organisms, including renal failure, gastrointestinal and central nervous system problems, dermatitis and lung cancer, have been reported [11]. Per the United States Environmental Protection Agency (USEPA), the maximum permitted concentration of Cr(VI) in drinking water is less than 0.1 mg/L, although the World Health Organization (WHO) recommends 0.05 mg/L [12]. USEPA classifies Cr(VI) as one of the 129 most critical pollutants [13].

The removal of hazardous chromium from residential and industrial water may be accomplished using a variety of physical and chemical processes, including coagulation, adsorption, ion exchange, chemical precipitation, electrochemical, and membrane techniques [14].

Finding adsorbents capable of removing Cr(VI) from aqueous environments is the focus of many extensive studies. Adsorbents for Cr(VI) removal have been produced and effectively utilized from a variety of carbonaceous materials, including biomass, graphene oxide, nanoparticles, and composite materials [10,15]. Biomass-derived carbonaceous compounds have lately been given more attention due to the exhibited high specific surface area, porosity, and rich surface functional groups [16]. The adsorption of Cr(VI) has been studied using activated carbon derived from seaweed [6,7], Bermuda grass [17], carbon-encapsulated hematite nanocubes [18], banana biochar [19], corn stalks [20], coffee waste [21,22], apple peels [23], Modified Spruce Sawdust [24], carbon nano-onions [25], magnetic nano-adsorbent impregnated with activated carbon [26], polymer-magnetic-algae nanocomposite [27], date palm empty fruit bunch wastes [28] and fox nutshells [29].

The adsorption for ionic contaminants in wastewater is limited by the number of hydrophobic surface functional groups on AC [30]. Hence, heteroatom dopant insertion has lately attracted more attention in water treatment applications because dopant atoms (such as N, F, S, P, and B) on the surface of activated carbon enhance the adsorption capacity of heavy metal ions [31,32,33]. There are many P-related surface groups and complexes on the AC surface that are thermally stable, making it suitable for use in water purification for Cr(VI) [34]. It is worth mentioning, from a chemical standpoint, that the introduction of an electronegative element (such as N, P, or O) into the carbon framework results in an increase in the positive charge density on adjacent carbon atoms, which promotes the adsorption of Cr(VI) ions. The use of H_3_PO_4_ as a chemical activator for (CO)_3_PO and (CO)_2_PO_2_, is known to increase thermal and textural stabilities [35].

On the other hand, the solute-solvent interactions at the molecular level can be studied using molecular dynamics (MD). This technique enables the examination of molecular areas with a finer size resolution than experimental methods [36]. According to earlier research, parameters such as solution pH and the presence of inorganic salts in the medium have an impact on the surface characteristics (zeta potential) of the activated carbon. By combining MD with surface potential data, it is possible to characterize the parameters that influence the adsorption of inorganic pollutants such as Cr(VI) on AC.

Despite the increasing number of studies dedicated to the development of adsorbent materials derived from biomass, only a few of them have been focused on Sargassum seaweed.

In this paper, a low-cost adsorbent derived from widely available pelagic Sargassum seaweeds was synthesized and studied to remove Cr(VI) chromium from polluted water. The structure and surface of the Sargassum seaweed were modified by a thermochemical activation process, converting the algae into activated carbon.

In order to comprehend the adsorption mechanism, the adsorption kinetics, isotherms, and thermodynamics were studied. Finally, MD simulations have been employed to understand the experimental findings and the solute-solvent interactions at the molecular level.

## 2. Materials and Methods

### 2.1. Materials

Phosphoric acid (H_3_PO_4_, 85%), potassium dichromate (K_2_Cr_2_O_7_, 98%), 1,5-diphenylcarbazide, sulfuric acid (H_2_SO_4_, 95–98%), hydrochloric acid (HCl, 37%) and caustic soda (NaOH, 98%), were purchased from Sigma Aldrich (France). Analytical grade magnesium chloride hexahydrate (MgCl_2_·6H_2_O) was purchased from VMR (France).

The seaweed biomass, *Sargassum* spp. (*Sargassum natans* and *Sargassum fluitans*) was harvested on Guadeloupe’s La Datcha beach.

Two AC were used in the adsorption investigations. A commercial AC from VEOLIA (Supercap BP10, SBP10) was used as a reference because of its large specific surface area and remarkable adsorption capacity, and a Sargassum AC named SAC600 3/1 was synthesized in the laboratory by phosphoric acid activation as described below.

### 2.2. Preparation of Sargassum AC

Tap water was used to rinse the Sargassum samples in order to eliminate any further algae, sand, or pollutants. After being cleaned with water, the Sargassum was dried in the sun for 7 days. Thereafter, it was placed in a standard oven set to 110 °C for 24 h to remove any remaining moisture. The dry sample was coarsely ground in an electric mixer-grinder and then sieved to acquire a size distribution ranging from 0.5 to 1.5 mm.

The sieved dried Sargassum was then impregnated in phosphoric acid using a mass ratio of 3/1 for 15 h before being pyrolyzed for 2 h at 600 °C using a Carbolite CTF 12/75/700 horizontal tubular furnace with a constant flow of N_2_ (80 mL/min) to minimize oxidation of the sample. The temperature was risen at 5 °C/min from room temperature up to the desired pyrolysis temperature.

Following pyrolysis, the cooled solid leftovers, unwashed AC, were crushed in a mortar to facilitate access to the pores. The Sargassum AC was then washed with deionized water (<5 µS  cm^−1^) in a beaker with stirring until a pH of 6.5–7 was achieved. To improve the washing of the AC and obtain a greater specific surface area, the samples were washed again with deionized water for 8 h, but this time in a stirred beaker at a higher temperature (80 °C), in a turbulent regime. This washing was repeated twice to ensure that the pores of the AC were completely free of phosphoric acid and other potential contaminants. Finally, the Sargassum AC was dried in an oven at 110 °C for 24 h to remove residual moisture. The yield of synthesized activated carbon was 14.5% (*w*/*w*%). This yield corresponds to values previously published where Sargassum algae were used as raw material for the production of activated carbon [7].

### 2.3. Adsorbents Characterisation

The specific surface area (S_BET_) and porous properties of each sample were determined using N_2_ adsorption studies. A Sorptomatic 1990 series analyzer was utilized to assess the N_2_ sorption capacity at 77 K. The samples were degassed at 250 °C for 17 h to eliminate residual moisture, prior to N_2_ adsorption studies. In the P/P_0_ range of 0.10321–0.225, the specific surface area was estimated using a linear fit of the Brunauer-Emmett-Teller (BET) equation. The volume of adsorbed N_2_ was defined as the total pore volume (V_Tot_) at a relative pressure (P/P_0_) of 0.9982. The Horvath-Kawazoe method was used to compute the micropore volume (V_micro_), and the mesopore volume (V_meso_) was calculated as V_meso_ = V_Tot_ − V_micro_ [7]. The ratio 4V_Tot_/S_BET_ was used to calculate the mean pore diameter (D_p_) [37].

The point of zero charge (pH_PZC_) or isoelectric point (pH_IEP_) is an essential feature that defines the linear range of pH sensitivity. It subsequently reveals the surface polarity that gives information on its adsorption capacity for ionic species [38]. The following procedure was used to estimate the AC samples point of zero charge (pH_pzc_). 0.1 g of carbon was added to 20 mL of 0.1 M NaCl solution at initial pH values ranging from 2 to 13 and adjusted with NaOH or HCl (0.5 M). The samples were agitated at a speed of 150 rpm for 24 h at a temperature of 25 °C. The initial (pH_i_) and final (pH_f_) pH values were determined and used to calculate the pH_pzc_.

Using the Boehm titration method, acidic and basic groups were determined. 500 mg of AC were added to 50 mL of NaOH 0.05M or HCl 0.05 M solution to determine the concentration of basic and acidic groups, respectively. On a shaker, the glass bottles were sealed and shaken for 48 h. The sample was then filtered with a 0.45 µm filter syringe to eliminate the AC. 10 mL of filtrate sample was titrated with HCl or NaOH (0.05 M). Every titration was performed in triplicate, and the average values are provided.

Attenuated Total Reflectance Fourier Transform Infrared Spectroscopy (ATR-FTIR) was used to identify the surface functional groups of both AC. The FTIR spectra were acquired at ambient temperature using the Spectrum Two instrument from PerkinElmer in the wavenumber range of 4000 to 650 cm^−1^ with a resolution of 2 cm^−1^.

A Kratos AXIS Ultra HSA X-ray Photoelectron Spectrometer coupled with a hemispherical electron analyzer was employed to conduct the X-Ray Photoelectron Spectroscopy (XPS) analysis that was required to determine the elements and oxidation states of the activated carbon surface. This analysis was carried out using a monochromatized Al K (1486.6 eV) X-ray excitation source. To deconvolute each spectrum, a Lorentzian function was applied.

### 2.4. Batch Adsorption Studies

The measurement of the concentration of chromium in solution was performed according to the standard colorimetric method of Greenberg [39]. A 1 mL chromium solution is sampled and mixed in an acid medium until reaching a pH of 2 ± 0.5 (3.3 mL of H_2_SO_4_ 0.1 M were used). Next, 1 mL of a complexing agent, 1,5-diphenylcarbazide (0.25 g/100 mL of absolute ethanol) was added to the mixture forming a purplish-pink complex. After 5 min, its concentration was measured at 540 nm using a SECOMAN Uviline 9400 spectrophotometer.

Dichromate concentrations varying from 0.5 to 7.5 mg/L (0.5, 1, 2, 3, 4 and 7.5) were used for the calibration curve. A stock chromium solution was prepared by dissolving a determined quantity of K_2_Cr_2_O_7_ in deionized water in order to obtain a concentration equal to 1000 mg/L. The other concentrations were obtained by successive dilutions.

The adsorption experiments (kinetic and isotherm studies) were conducted by contacting 0.1 g of AC with 100 mL of 60 mg/L of chromium solution. The experiments were conducted in a thermostatically controlled water bath at 30 °C and 150 oscillations per minute.

To determine the equilibrium, the sampling of the aqueous phase was performed at predetermined time intervals. Each sample was filtered with a Nylon syringe filter of 0.2 µm, and then, analyzed by UV-visible spectrophotometry according to the previously described conditions. The initial pH of the chromium dissolution was measured (pH = 5.0 for the dissolution of 60 mg/L of K_2_CrO_7_, for example).

The duration of the experiment and the time required to reach the plateau of the adsorption kinetics are directly related to the characteristics of the precursor. Normally, 24 h is enough time to reach the plateau for the AC sample, but it is advisable to continue the measurements until at least 48 h.

To assess the effect of contact time, sampling was conducted every 15 min for the first 2.5 h, and subsequently at larger intervals until 48 h. The standard colorimetric method of Greenberg was used to calculate the Cr(VI) equilibrium concentration, *C_t_* (mg/g). Equation (1) computes the amount of Cr(VI) adsorbed under equilibrium at each time, *q_t_*.


*Equation (1) Adsorption capacity at the time “t”.*




(1)
$$q_{t}=\frac{C_{0}-C_{t}}{m}\times V$$



Several kinetic models were studied to investigate the adsorption mechanism.

To obtain adsorption isotherms, several conditions at initial concentrations of 10; 15; 20; 25; 30; 35; 40; 45; 55; 60; 75; 100 mg/L of chromium solution were prepared. The pH was measured for each of the prepared solutions. As for adsorption kinetics, the tests were carried out in a thermostatic water bath at 30 °C and stirred at 150 oscillations per minute.

A series of experiments were conducted to investigate the adsorption of Cr(VI) ions onto activated carbons and the influence of various parameters. Equations (2) and (3) were used to acquire the equilibrium adsorption capacity, *q_e_* (mg/g), and the hexavalent chromium removal percentage [40].


*Equation (2) Adsorption capacity at equilibrium.*




(2)
$$q_{e}=\frac{C_{0}-C_{e}}{m}\times V$$




*Equation (3) Cr^6+^ removal percentage.*




(3)
$$Removal\;\left(\%\right)=\frac{C_{0}-C_{e}}{C_{0}}\times 100$$



C_0_ and C_e_ (mg/L) are the initial and final concentrations of Cr^6+^ ions. V is the volume of the solution (L) and m is the mass of the adsorbent (g).

To study and fit the experimental data, six non-linear regression models were used. Langmuir and Jovanovic models normally show a good fitting a homogeneous surface without lateral interactions is studied (Equations (4) and (5)). Fowler model is used for homogeneous surfaces with lateral interactions (Equation (6)). The Freundlich and Jovanovic-Freundlich models fit well when the surface is heterogeneous and there are not lateral interactions (Equations (7) and (8)). Finally, heterogeneous surfaces with lateral interactions can be described by the Fowler–Guggenheim/Jovanovic–Freundlich model (Equation (9)) [41].


*Equation (4) Langmuir adsorption capacity at equilibrium.*




(4)
$$q_{e}=\frac{q_{s}\cdot \;K\cdot \;C_{e}}{1+K\cdot C_{e}}$$




*Equation (5) Jovanovic adsorption capacity at equilibrium.*




(5)
$$q_{e}=q_{s\cdot }\left(1-e^{-\left(K\cdot \;C_{e}\right)}\right)$$




*Equation (6) Fowler adsorption capacity at equilibrium.*




(6)
qe=qs· K· Cee−χ· qeqs+K· Ce




*Equation (7) Freundlich adsorption capacity at equilibrium.*




(7)
$$q_{e}=K\cdot \;C_{e}^{\left(\nu \right)}$$




*Equation (8) Jovanovic–Freundlich adsorption capacity at equilibrium*




(8)
$$q_{e}=q_{s\cdot }\left(1-e^{-{\left(K\cdot \;C_{e}\right)}^{}}\right)$$




*Equation (9) Fowler–Guggenheim/Jovanovic–Freundlich adsorption capacity at equilibrium.*




(9)
qe=qs·1−e−K· Ce·eχ· qeqs



Adsorption capacity and solution concentration at equilibrium are represented by *q_e_* (mg/g) and *C_e_* (mg/L), respectively. *q_s_* is the monolayer capacity, χ is the adsorbate–adsorbate interaction parameter, ν is the heterogeneity parameter and K are the affinity parameters of each model.

Using a modified Newton algorithm, regressions of the experimental data to the adsorption iso-therm models were conducted. By minimizing the residual sum of squares (RSS), this technique de-termines the values of the isotherm parameters.


*Equation (10) Residual sum of squares (RSS).*




(10)
$$RSS=\overset{n}{\underset{i=1}{\sum }}{\left(q_{exp,\;i}-q_{t,\;i}\right)}^{2}$$



The applicability of the kinetic and isotherm models was assessed by evaluating the coefficient of determination, R^2^.

Several investigations were also conducted to find the optimal conditions for the adsorption of hexavalent chromium with AC derived from Sargassum seaweed.

The impact of pH on chromium adsorption was also examined. HCl and NaOH were used to alter the pH of the chromium solution to a range of 2 to 11.

The effect of the solution temperature was also studied at 25 °C, 30 °C and 35 °C for a 6 mg/L Cr(VI) solution. In these investigations, a solution of 100 mL containing 6 mg/L of K_2_CrO_7_ and 10 mg of activated carbon was utilized. Using the Van’t Hoff equation, several thermodynamic parameters were determined based on the outcomes of the thermal interaction. Using Equations (11)–(13), the standard Gibbs free energy change (ΔG), standard enthalpy change (ΔH), and standard entropy change (ΔS) were calculated [42].


*Equation (11) Equilibrium constant.*




(11)
$$K_{eq}=\frac{C_{ad}}{C_{e}}=\frac{C_{0}-C_{e}}{C_{e}}$$




*Equation (12) Standard Gibbs free energy.*




(12)
ΔG=ΔH−T·ΔS




*Equation (13) Van’t Hoff equation.*


(13)$$lnK_{eq}=\frac{\Delta S}{R}-\frac{\Delta H}{R}\cdot \frac{1}{T}$$
where *K_eq_* is the equilibrium constant, *C_ad_* is the concentration (mg/L) of adsorbed Cr^6+^ onto the AC, *C_e_* is the concentration (mg/L) in solution at equilibrium, *C*_0_ is the initial concentration (mg/L) of the solution, R is the universal gas constant (8.314 J/(mol·K)), *K_eq_* is the equilibrium constant.

Finally, the effect of ionic strength and counter ions on the adsorption of hexavalent chromium adsorption was investigated. The purpose of this experiment was to confirm the results obtained during molecular dynamics simulations. In this experiment, 4.3 mg of MgCl_2_ were diluted in 10 mL of deionized water. For 2 h, the magnesium solution was mixed with 10 mg of activated carbon. Following that, 20 mL of 30 mg/L chromium solution were added. The final mixture was shaken at 150 oscillations per minute for 24 h before the absorbance of the resulting solution was measured.

All the experiments were carried out at least 3 times per sample evaluating experimental reproducibility.

### 2.5. Molecular Dynamics Simulations

To understand the adsorption of ions and investigate the interactions between inorganic micro-pollutants and the AC surface at the molecular level, and interpret the experimental data of hexavalent chromium adsorption at the surface of porous carbons, classical MD simulations were used. For the design of the activated carbon, the ratios oxygen/carbon from the XPS analyses of SAC600 3/1 were used to estimate the proportion of functional groups. O-C=O groups revealed the lowest proportion (4.24%). To limit the size of the molecule and ease the simulations, it was considered that the 4.24% of O-C=O represented 2 carboxylic groups. Subsequently, for C=O groups, the measured 7.43% was considered as 4 carbonyl groups. Finally, the 21.04%, derived from the C-O functional groups, was interpreted as 8 hydroxyl groups. Therefore, SAC600 3/1 model was developed having 23 rings, 2 carboxyl, 4 ketone, and 8 hydroxyl groups.

Figure 1 shows the design of the SAC600 3/1 and a snapshot of the simulation.

This model (Figure 1a) can be used in a pH interval from 4 to 8. The model was designed to work under real water conditions of drinking water and wastewater (pH around 7). For this reason, carboxyl groups are deprotonated (-COO^−^, pK_a_ = 3 to 6) and phenol groups are protonated (-OH, pK_a_ = 8 to 11) [43,44]. As carboxyl groups are deprotonated therefore for the negatively charged model of activated carbon, a partial repulsion of chromium anions is expected. Density functional theory (DFT) was used for deriving the Cr(VI) ion parameters which were then employed in the molecular dynamics simulations. In DFT calculations, an implicit solvation model was applied. However, explicit solvation was used in the productive classical MD simulations. Density functional theory calculations have been presented in Appendix A. The force field parameters for Cr^VI^ ions and the activated carbon molecule were described by the potential function based on General Amber Force Field (GAFF) parameters [45,46]. On the other hand, for Na^+^, Cl^−^ and Mg^2+^ ions the ff99SB [47] amber force field, which is compatible with GAFF, has been applied.

To prepare the simulation boxes, one molecule of AC model and a different ratio of HCrO_4_^−^/CrO_4_^2−^ depending on the pH of the solutions, were added to the simulation boxes. Moreover, to increase the number of positive binding sites and to study the effect of ionic strength and counter ions on the adsorption of hexavalent chromium, magnesium chloride has been added to the simulation boxes to assess the impact of magnesium ions on the adsorption of Cr^6+^ on AC. As the AC model surface was negatively charged, K^+^ ions were added to the system as counterions to compensate the charge, but also to simulate the experimental conditions as chromium solutions were prepared from K_2_CrO_7_.

Figure 1b shows a snapshot from the MD simulation where the Mg^2+^ ions are added to the solution. For performing the simulations, first of all, systems were minimized by steepest descent minimization then equilibrated by 500 ps NVT (Canonical ensemble) followed by 500 ps NPT (isothermal–isobaric ensemble). The linear constraint solver (LINCS) algorithm [48] was used for equilibration and short-range non-bonded interactions were cut-off by 1.2 nm. The particle mesh Ewald method [49] was used for long-range electrostatic interactions. All simulations were run at 300 K with a V-rescale coupling algorithm [50] with a coupling constant of 0.1 ps. Production runs were performed at NPT ensemble for 100 ns with 2 fs time steps. Gromacs 4.6.5 program package was used for performing MD simulations [51]. For visualizations, Visual Molecular Dynamics (VMD) was applied [52].

Atom types, atomic point charges and indicated inter-atomic bonding in CrO_4_^2^^−^, HCrO_4_^−^ and Cr_2_O_7_^2^^−^ anions can be found in the Appendix A.

## 3. Results and Discussion

### 3.1. Textural Characterization

#### Surface Area and Pore Size Distribution

The robust nature and porous structure of AC produced from *Sargassum* seaweed satisfied the requirements for gas sorption experiments. Studies on gas adsorption were conducted on the SAC600 3/1 and SBP10 up to a relative pressure (P/P_0_) of 0.9982. As demonstrated in Figure 2, the sorption of nitrogen at 77 K exhibited gradual adsorption.

The SBP10 structure resembles a type I isotherm, indicating a greater presence of micropores. SAC600 3/1 displayed a type II isotherm with considerable adsorption below P/P_0_ = 0.05 and an H4 hysteresis loop at P/P_0_ 0.48. Typically, type II isotherms are observed in hierarchical porous solids with a broad range of pore-size distributions spanning the micro, meso and macropore regimes, and where the corresponding porosity varies substantially in each zone [53]. The H4 hysteresis loop’s adsorption branch is a mix of types I and II, with more prominent adsorption at low P/P_0_ being linked to micropore filling. H4 loops are generally abundant in micro-mesoporous carbons [54].

SAC600 3/1 and SBP10 revealed BET surface areas of 1695 m^2^/g and 1900 m^2^/g, respectively. SAC600 3/1 exhibited a very large specific surface area, making it a highly valuable AC. Its specific surface area surpasses not just activated carbons obtained from Sargassum but also the vast majority of porous carbon compounds derived from seaweed [6,7,8,55,56]. In addition, it is competitive with the most recently published porous carbon compounds derived from biomass [57,58,59,60].

The presence of micropores and mesopores in the material was confirmed by the HK (Horváth-Kawazoe) pore size distribution pattern [61]. Figure 3 shows the corresponding results.

Based on the pore size distribution calculated from the adsorption isotherms using the HK model, the corresponding average pore sizes of SAC600 3/1 and SBP10 were 0.6 nm and 0.8 nm. Table 1 summarizes the textural properties of both activated carbons.

The proportion of micropores to mesopores differed among the studied activated carbons. SAC600 3/1 contained 40% of micropores and 60% of mesopores. In contrast, the results for SBP10 revealed the exact opposite, with 60% micropores and 40% mesopores.

### 3.2. Chemical Characterization

#### 3.2.1. pH_PZC_

The pH at the point of zero charge (pH_PZC_) was determined for SAC600 3/1 and SBP10. It is essential to estimate the pH_PZC_ of the AC since the anion adsorption is favored when pH of the adsorbate solution is lower than the pH_PZC_ value, as the surface of the porous material is charged positively. In contrast, if the pH is greater than the pH_PZC_ value, the surface will be charged negatively, hence increasing the cations’ adsorption attraction. The pH_PZC_ results for SAC600 3/1 and SBP10 were similar, 3.40 to 3.70, respectively.

#### 3.2.2. Boehm Analysis

The surface acidity and basicity of both activated carbons were determined by the Boehm titration technique. This method provides quantitative measurements for oxygen-containing surface functional groups on carbon materials, supposing that carboxyl, phenolic, and lactone groups are reduced by a sodium hydroxide solution and basic groups are neutralized with a hydrochloric acid solution. The results for each activated carbon are presented in Table 2.

The number of acidic functional groups is higher than the number of basic functional groups for both activated carbons. The number of acid groups in SAC600 3/1 is double that in SBP10, which confirms that the acid activation process with phosphoric acid, used to obtain activated carbon from Sargassum, guarantees the presence of more acid groups on the surface of the adsorbent. On the other hand, this result justifies the selection of the activated carbon model for MD simulations, where mainly acid groups were considered.

#### 3.2.3. FTIR Analysis

FTIR spectroscopy was used to identify the chemical surface groups of the carbon materials under investigation. Figure 4 depicts the FTIR spectra generated for SAC600 3/1 and SBP10.

Table 3 comprises columns indicating the possible structure of the group, the wavenumber range in which it is most frequently observed, the exact position of the measured peak, and its probable designation.

The principal peaks of the spectra for both activated carbons are comparable (detection at 1569 cm^−1^ and 1081 cm^−1^). Commonly, these peaks correspond to the skeletal C=C vibrations of aromatic rings and alcohol functional groups, respectively. Additionally, SAC600 3/1 displays a significant peak at approximately 1184 cm^−1^ confirming one more time, the efficiency of preparing AC by phosphoric acid activation. Nevertheless, because the detected peaks are so large, it is difficult to confirm the presence of other functional groups. Both activated carbons are likely to contain functional groups including alcohols, carboxylic, ester, and/or phosphorus-based groups. Some nitrogenous compounds may be present as well.

Around 3400 cm^−1^ to 2800 cm^−1^, hydroxyl groups can be detected. Depending on the sample, these peaks are easy or hard to identify. The presence of hydroxyl groups is expected taking into account that the material has been activated with H_3_PO_4_. Furthermore, on activated carbons derived from seaweed, the modest detection of asymmetric and symmetric CH_2_ stretching vibrations about 2900 cm^−1^ (usually with two peaks) has already been reported.

#### 3.2.4. XPS Analysis

XPS spectroscopy was employed to characterize the chemical elements and functional groups present on the surface of SAC600 3/1 and SBP10. Table 4 summarizes the data gathered from the XPS analysis.

The primary elements detected in the SAC600 3/1 were C, O, N, S, and P. The presence of nitrogen and phosphorus seems normal considering that Sargassum seaweeds also have these elements in their composition [6,64]. Furthermore, the AC was chemically activated with H_3_PO_4_ which is also a source of phosphorus and oxygen. The absence of Na, Cl, Si and Ca, confirms the good development of the washing procedure where these atoms were successfully removed. The contents of carbon C1s and oxygen O1s were 83.30% and 13.14%, respectively, confirming the formation of a carbonaceous matrix. SBP10 showed a higher content of C 1s (92.99%). It is very possible that the commercial AC was prepared at a higher temperature as the carbon content is higher compared to the other atoms.

Four separate elements with distinct binding energies were discovered in the C1 s spectra. The C 1s deconvolution (C-C, C-H, and C=C interactions) accounted for 67.23% of the atomic concentration that was measured. The highest values were seen in C=C interactions. The observation of a partially graphitic surface supports the FTIR analysis conclusions. Similar to this, the FTIR analysis also registered the presence of C-O compounds (21.09 %), indicating the presence of a large number of phenolic groups or compounds with similar structures. Finally, it was discovered that relative atomic concentrations of carbonyl (C=O) and carboxyl (O-C=O) groups showed lower values (7.43% and 4.24%, respectively). The deconvoluted carbon spectrum of the SBP10 revealed a high concentration of C-C, C-H and C=C bonds (80.88%), where 57.72% corresponded to C=C interactions, indicating the formation of a significant graphitic surface. Additionally, a considerable number of C-C and C-H groups were identified. Compared to the SAC600 3/1 values, the C-O groups of SBP10 are less prevalent (7.45%). This may have been induced by the pyrolysis temperature employed during its production.

### 3.3. Adsorption Experiments

#### 3.3.1. Influence of Different Parameters on Chromium Adsorption

The impact of the following parameters was investigated: solution pH, temperature, solution ionic strength with MgCl_2_ and contact time (adsorption kinetics). Figure 5 displays the charts obtained from the different studies.

#### 3.3.2. Influence of Solution pH

The effect of pH on the removal of Cr(VI) by adsorption on SAC600 3/1 was investigated by altering the pH values (2–11) and allowing 24 h of contact at 30 °C for each pH value. The outcomes are depicted in Figure 5a. The pH had a considerable effect on the adsorption of Cr(VI), as indicated by the abrupt fluctuations in the adsorption capacity as a function of the pH of the solutions. The pH altered the adsorption behavior by dissociating functional groups on the active sites of the adsorbent’s surface [65] at a high pH value. The optimal pH value for the adsorption of Cr(VI) was estimated to be roughly 2, at which Cr(VI) removal yield was approximately 100%. The adsorption levels of Cr(VI) reduced dramatically as the pH climbed from 2 to 5. Above pH 9, adsorption values are negligible. Numerous investigations [65,66,67] have revealed similar results for Cr(VI) adsorption by various lignocellulose-based materials.

As a function of pH, the Cr(VI) species can exist in many forms such as H_2_CrO_4_, HCrO_4_^−^, CrO_2_^−^, and Cr_2_O_7_^2−^ in the solution phase. The form of Cr(VI) is regulated by the pH of the solution according to equilibrium reactions listed below [29]:
$$H_{2}CrO_{4}↔HCrO_{4}^{-}+H^{+}\;pKa=4.1$$
$$HCrO_{4}^{-}↔CrO_{4}^{2-}+H^{+}\;pKa=5.9$$
$$CrO_{7}^{2-}+H_{2}O↔2HCrO_{4}^{-}\;pKa=2.2$$

In highly acidic solutions, it is known that HCrO_4_^−^ and Cr_2_O_7_^2^^−^ ions prevail among Cr(VI) species [68]. Hydronium ions that surround the surface of the adsorbent at extremely low pH levels assist Cr(VI) in attaching to the adsorbent’s binding sites due to their increased force of attraction. As the pH increased, the overall charge on the surface of the adsorbent became negative, resulting in a decrease in the adsorption of Cr(VI) oxyanions [69].

For this reason, carboxyl groups are deprotonated (-COO^−^, pK_a_ = 3 to 6) and phenol groups are protonated (-OH, pK_a_ = 8 to 11) [43,44]. As carboxyl groups are deprotonated and therefore, negatively charged, a partial repulsion of chromium anions is expected.

On the other hand, at pH ≈ 5–7, which is most likely the case for drinking water, a rough estimation gives deprotonation of OH surface groups lower than 3% and can be neglected, while only COOH groups are deprotonated to a considerable extent (≈90%) [70].

At pH below 4, the acid groups are protonated, and the AC surface is more positively charged than when the pKa of the carboxyl and hydroxyl groups are reached. From pH 4 to 8, almost all of the carboxyl groups are deprotonated, while only about 3% of the hydroxyls are deprotonated [70]. Above pH 8, almost all hydroxyls are already deprotonated [43,44]. This explains why there is high adsorption at pH 2, medium at 5, and very poor thereafter.

Although the adsorption capacity is higher at pH 2, the remaining studies were conducted at pH 5 since the pH range between 5 and 7 better reflects the pH of drinking water and wastewater.

#### 3.3.3. Influence of Solution Temperature and Thermodynamic Studies

Temperature is a crucial parameter in adsorption processes, as it determines whether the adsorption is exothermic or endothermic. Figure 5b illustrates the results of these experiments. Cr^6+^ adsorption decreased with temperature and reached its minimum value at 35 °C, indicating that Cr(VI) ion adsorption on SAC600 3/1 is an exothermic process and physisorption is the main adsorption mechanism. Furthermore, this conclusion has also been reported by other researchers [71,72,73,74]. When the temperature of the solution was increased from 25 to 35 degrees Celsius, the Cr^6+^ adsorption percentage decreased from around 66% to 55% for the studied solution. Temperature plays a crucial role in the chemical interactions between the surface groups of an adsorbent and the ions that it adsorbs. Therefore, the hexavalent chromium adsorption capacity is a variable that depends on temperature.

The thermodynamic characteristics are essential for comprehending adsorption mechanisms since they demonstrate the process’s feasibility, spontaneity, and heat transfer [7].

The thermodynamic studies were conducted at 25 °C, 35 °C and 40 °C. The Vant Hoff equation has been utilized to compute Gibbs free energy, standard enthalpy change, and standard entropy change.

ΔH and ΔS were determined from the plot of ln K_eq_ as a function of 1/T (Appendix A). The thermodynamic characteristics of adsorption are listed in Table 5. The negative value of ΔG reflects that chromium interactions with SAC600 3/1 are spontaneous at lower temperatures and less spontaneous at higher temperatures. The negative value of ΔH indicates the exothermic nature of the adsorption process in the studied temperature range [33]. The negative ΔS value denotes the affinity of SAC600 3/1 for Cr(VI) ions with less randomness at the adsorbent solid-solution interface during the adsorption of Cr(VI) [75].

#### 3.3.4. Influence of AC Pre-Treatment with Magnesium Chloride

The effect of magnesium ions on the surface and pores of SAC600 3/1 was investigated for its prospective application as a promoter of hexavalent chromium adsorption. In the present study, Sargassum AC impregnated with an aqueous magnesium chloride solution was utilized and compared to non-modified SAC600 3/1 for the sorption of Cr^6+^.

Figure 5c demonstrates that impregnating SAC600 3/1 with Mg^2+^ ions before adsorption, increased the adsorption capacity of the adsorbent by 5.6%. The efficient loading of magnesium ions on the adsorbent surface might give more sorption sites. Mg^2+^ ions have a very small radius of 79 pm, which enables them to penetrate all the cavities of the activated carbon, embedding themselves in the walls of the micropores and mesopores. This allows the AC surface to be more positively charged, resulting in a higher affinity for anions pollutants in aqueous solution [76,77], and so, promoting the attraction of chromium anions such as HCrO_4_^−^ and Cr_2_O_7_^2^^−^.

#### 3.3.5. Adsorption Kinetics

The uptake of chromium from solution by SAC600 3/1 increased with contact time and attained equilibrium after 24 h at 30 °C. However, 64% of the Cr^6+^ adsorption occurred in the first 15 min of contact in a very fast adsorption process. After 30 min, the removal percentage was 73%. This high adsorption speed is justified by the large specific surface area of both activated carbons and the presence of hydrophilic functional groups. The results of these experiments are shown in Figure 5d.

After one hour of contact, both activated carbons had comparable outcomes. However, after the first hour, SAC600 3/1 showed a greater adsorption capability. The FTIR and XPS analyses revealed that SAC600 3/1 contained more oxygenated groups, which could improve the electrostatic attraction, hence promoting the adsorption of dichromate, presumably, by electrostatic interactions of the Van der Waals type.

Four kinetic models, pseudo-first-order, pseudo-second-order type 1, pseudo-second-order type 2, and intraparticle diffusion model, were employed to analyze the adsorption mechanism [78]. The following equations represent the different models [79]:


*Equation (14) Pseudo-first order.*




(14)
$$\log\left(q_{e}-q_{t}\right)=log(q_{e})-\frac{K_{1}}{2.303}\;\cdot \;t$$




*Equation (15) Pseudo-second order Type 1.*

(15)
$$\frac{1}{q_{t}}=\frac{1}{K_{2}\;\cdot \;q_{e}^{2}}+\frac{t}{q_{e}}$$




*Equation (16) Pseudo-second order Type 2.*

(16)
$$\frac{1}{q_{t}}=\left(\frac{1}{K_{2}\;\cdot \;q_{e}^{2}}\right)\;\cdot \;\frac{1}{t}+\frac{t}{q_{e}}$$



*Equation (17) Intraparticle diffusion model.*(17)$$q_{t}=k_{ip}\;\cdot \;t^{0.5}+C_{i}$$
where *q_t_* is the amount of adsorbate adsorbed at time *t* (mg·g^−1^), *q_e_* is the amount of adsorbate adsorbed at equilibrium (mg·g^−1^), *t* is the time (min), *k_1_* is the rate constant of the pseudo-first-order kinetics (min^–1^), *k_2_* is the rate constant of the pseudo-second-order kinetics (g·mg^−1^·min^−1^), *k_ip_* the rate constant for intraparticle diffusion (mg·g^−1^·min^−0.5^) and *C_i_* (mg·g^−1^) is constant and gives information about the thickness of the boundary layer: A larger value of *C_i_* suggests high boundary layer effect.

Table 6 summarizes the results obtained from the different models.

The best-fit model for SAC600 3/1 and SBP10 was the pseudo-second-order model, which had a correlation coefficient of 0.9973 and 0.9999, respectively. The pseudo-second-order equation describes the solid-liquid reaction and their adsorption kinetics mechanism, demonstrating that available active sites play a more important role in the adsorption process rather than Cr(VI) concentration [78,80]. Furthermore, the rate of a chemical reaction influences the availability of active sites on the surface of porous carbon materials [81,82,83,84]. Therefore, this model also indicated that chemisorption can partially influence the rate-controlling step in the adsorption processes. Several researchers have observed comparable results when utilizing biosorbents, biochars and activated carbons for the removal of Cr^6+^ [85,86,87].

The inapplicability of the intra-particle diffusion model further suggested that the rate of Cr(VI) ion adsorption was not driven by pore or intraparticle diffusion [88,89,90].

In conclusion, it is quite likely that total sorption is a combination of physisorption and chemisorption, where physisorption is the main process.

#### 3.3.6. Adsorption Isotherms

Langmuir, Jovanovic, Fowler, Freundlich, Jovanovic-Freundlich and Fowler-Guggenheim/Langmuir-Freundlich nonlinear models were used to match the experimental adsorption isotherms. The calculations were performed for both activated carbons, SAC600 3/1 and SBP10. The experiments were conducted at 30 °C with an initial pH of 5.0.

Table 7 summarizes the calculated parameters of the six models.

The Jovanovic-Freundlich isotherm was found to be the best fit for Cr(VI) adsorption onto SAC600 3/1, describing the heterogeneous adsorption with no lateral interactions [41].The Langmuir model estimated a maximum adsorption capacity of 54.54 mg/g with a correlation coefficient (R^2^) value of 0.9728 providing a more accurate description of the experimental data. SAC600 3/1, which was synthesized from Sargassum seaweed, demonstrated competitive results with similar published studies [63,75].

On the other hand, the commercial AC (SBP10) revealed that the Freundlich model fitted well with the experimental parameters, having a R^2^ of 0.9749, which explains the non-linear monolayer and heterogeneous chemisorption [91]. This suggests that Cr(VI) adsorption is enhanced on heterogeneous surfaces, resulting in multilayer adsorption at the binding sites between the surface functional groups of SBP10 and Cr(VI) ions [92]. A maximum adsorption capacity of 45.88 mg/g was obtained for SBP10.

The surfaces of the examined activated carbons are highly heterogeneous. This is mainly due to their pore size distribution and the presence of functional groups on the AC surface. Despite that the Langmuir equation is still frequently employed to evaluate adsorption isotherms on ACs, it is unsurprising that frequently used models similar to Langmuir and Jovanovic do not fit the data well for the examined ACs as these models are more representative of homogeneous surfaces.

The best models for each AC and the experimental are shown Figure 6.

Both AC, SAC600 3/1 and SBP10, have comparable adsorption capacities. Although the adsorption of SBP10 is faster initially, the maximum adsorption capacities of both AC are similar. Despite the higher specific surface area of SBP10, the maximum adsorption capacity of SAC600 3/1 is slightly greater. This is potentially due to the higher pore volume of SAC600 3/1.

### 3.4. Molecular Dynamics Simulations

To support the experimental results of Cr^6+^ adsorption on activated carbon at the molecular level, classical MD simulations were performed, and trajectories were analyzed. In order to quantify the adsorption of different species of hexavalent chromium ions in the simulations, radial distribution function (RDF), denoted as g(r), is used. The radial distribution function describes the probability distribution of finding a particle in the distance of r from another particle. This property can be calculated from the atomic positions of MD simulation by the analysis of trajectory. The radial distribution function of K^+^, HCrO_4_^−^ and CrO_4_^2^^−^ ions around activated carbon molecule is depicted in Figure 7.

The radial distribution functions show sharp peaks at the distance (r) of around 0.2 nm for K^+^ ions from the surface of activated carbon which proves that the K^+^ ions are adsorbed by carboxylate groups of activated carbon. By closer investigation of RDF, it can be observed that the peak intensity of K^+^ ions decreased when the pH of the solution increased from 5 to 7 which supports the idea that the K^+^ ions in the solution have less attraction to the surface of AC as the number of doubly charged CrO_4_^2−^ is increased in the solutions, therefore, K^+^ ions tend to make contact ion pair with CrO_4_^2−^ ions. The same trend for a peak intensity of K^+^ at the distance of about 1 nm also is visible which shows that a similar phenomenon is happening on the adsorption of K^+^ to the surface of activated carbon. Moreover, RDF plots show that the hexavalent chromium ions namely HCrO_4_^−^ and CrO_4_^2−^ adsorption decreased when the pH of the solution is increased from 5 to 7 which is in agreement with experimental results. The reason why the increase in pH decreases the adsorption of hexavalent chromium ions at the surface of activated carbon is due to the fact that the ratio of HCrO_4_^−^/CrO_4_^2−^ decreases therefore the hydrogen bonds of HCrO_4_^−^ with carboxylate groups (-COO^−^) is disrupted and only CrO_4_^2−^ ions can interact with -OH groups of the surface.

In addition, the experimental results showed that the impregnation of activated carbon with Mg^2+^ ions prior to the adsorption experiment, improves the adsorption of hexavalent chromium at the surface of AC. Therefore, MD simulations have been performed for the same system to understand the mechanism of the adsorption process. Radial distribution functions of the systems containing MgCl_2_ are shown in Figure 8 where the peak intensities for both HCrO_4_^−^ and CrO_4_^2−^ anions slightly increased which is in agreement with experimental results.

The atomic type, bond type, valence-angle type and dihedral-angle type parameters of the previous DM simulations can be found in the Appendix A. A comparison of vibrational frequencies (cm^−1^) of vacuum Cr^6+^ anions as calculated in DFT (PBE0/Aug-CC-pVDZ) and in force field (FF) using the fitted parameters can be found in Appendix A.

## 4. Conclusions

The valorization of invasive Sargassum seaweed as a source material for the production of AC was assessed in this paper. A Sargassum AC was synthesized at 600 °C using a chemical impregnation mass ratio of 3/1 (H_3_PO_4_ 85%/dried Sargassum seaweed). A high specific surface area of 1695 m^2^/g and a highly fictionalized surface were found for SAC600 3/1 indicating the potential use of Sargassum sp. as an interesting precursor for multiple applications. The pH of the solution was identified as the most important parameter for the adsorption performance. In a very acidic medium (pH around 2 or below), the Cr(VI) removal percentage was around 100%. On the other hand, the increase of solution pH dramatically reduces the adsorption capacity of the studied AC. Indeed, the removal percentage of Cr^6+^ at pH 9 was almost negligible. In the same way, the increase in the solution temperature, from 25 °C to 35 °C, slightly reduced the chromium uptake indicating the existence of an exothermic process. To improve the chromium adsorption capacity, magnesium chloride was employed as an activated carbon surface modifier resulting in a 5% increase compared to the AC without magnesium chloride. The adsorption behavior of Cr^6+^ on SAC600 3/1 and SBP10 may be described using a type 1 pseudo-second-order kinetic model. Moreover, the Jovanovic-Freundlich and the Freundlich adsorption isotherm were more appropriate to represent the Cr^6+^ adsorption mechanism onto both adsorbents, SAC600 3/1 and SBP10, respectively. The molecular dynamics simulations confirmed the experimental findings. Thus, the increase of solution pH causes a decrease in adsorption of hexavalent chromium anions on the surface of the AC. MD simulations and in particular, radial distribution functions (RDFs) analysis indicated that the hydroxyl and carboxylate groups of the activated carbon are the most influential functional groups for the interplay of solute-solvent interaction in the adsorption process of chromium anions. Moreover, MD simulations confirmed experimental results showing that the addition of MgCl_2_ to the surface of SAC600 3/1 increases the adsorption of anions. In conclusion, SAC600 3/1 can be potentially used as an effective and environmentally acceptable adsorbent to enhance the removal of Cr^6+^ from aqueous solution, and it could also provide a theoretical foundation for the removal of toxic heavy metals from water.

## Figures and Tables

**Figure 1 molecules-27-06040-f001:**
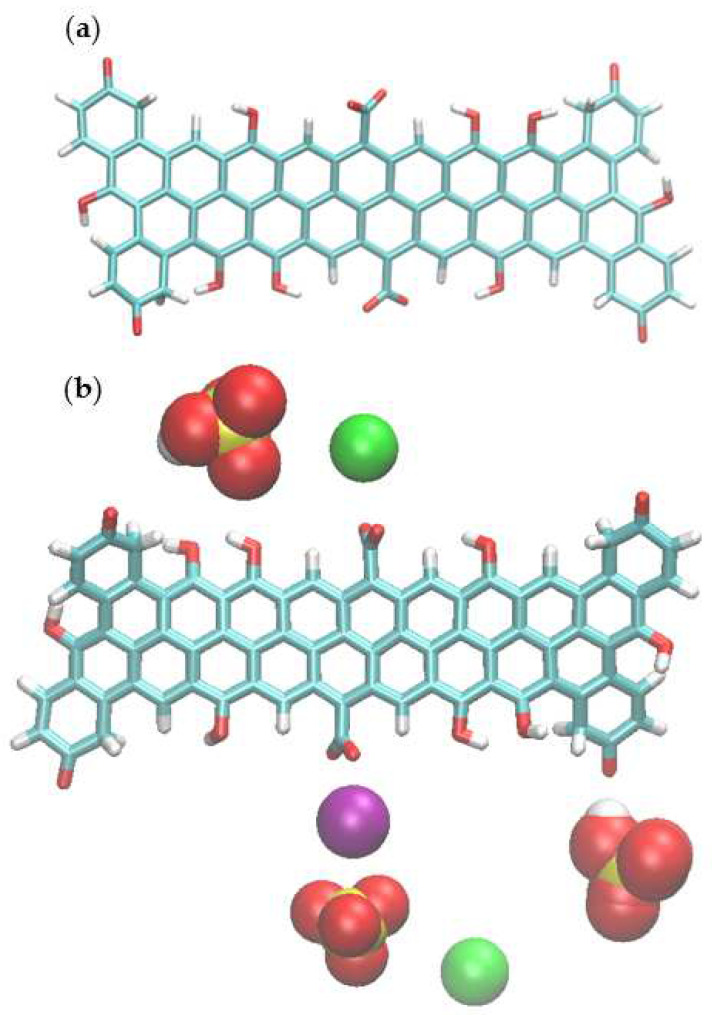
Proposed simplified surface model of (**a**) SAC600 3/1 (**b**) and ions around model activated carbon in the solution of activated carbon HCrO_4_^-^, CrO_4_^2-^ and MgCl_2_. Colour coding: carbon cyan, oxygen red, potassium ions (K^+^) green, magnesium ion (Mg^2+^) purple, hydrogen white and chromium yellow.

**Figure 2 molecules-27-06040-f002:**
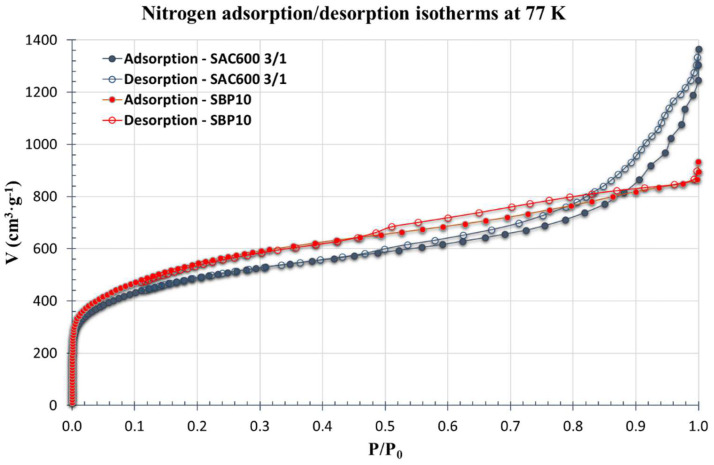
Nitrogen adsorption/desorption isotherms at 77 K on SAC600 3/1 and SBP10.

**Figure 3 molecules-27-06040-f003:**
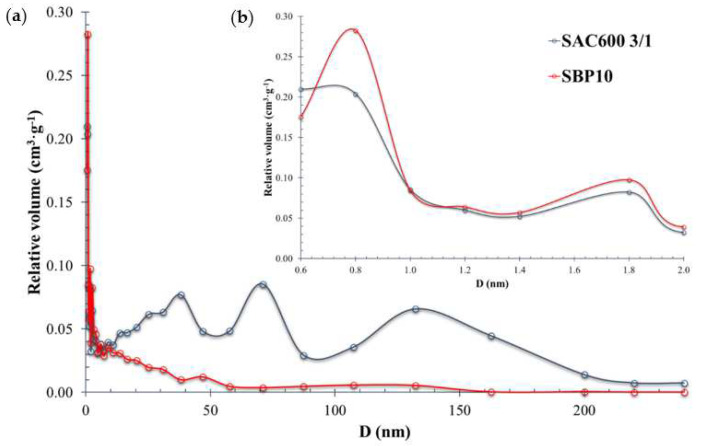
(**a**) Pore size distribution using HK method for both AC. D values between 0 and 240 nm. (**b**) Micropore analysis: D values between 0.6 and 2.0 nm.

**Figure 4 molecules-27-06040-f004:**
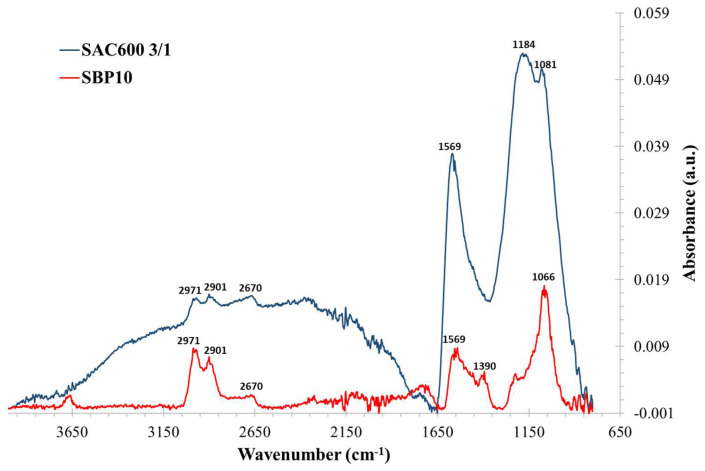
FTIR spectra of Sargassum based activated carbons at lab and pilot conditions.

**Figure 5 molecules-27-06040-f005:**
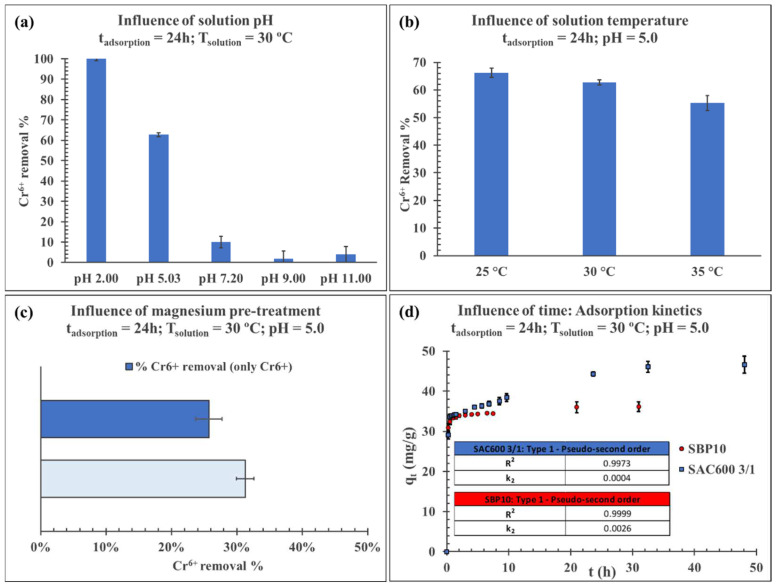
Adsorption experiments: (**a**) Influence of solution pH; (**b**) influence of solution temperature; (**c**) influence of solution ionic strength with magnesium chloride; (**d**) influence of contact time (adsorption kinetics).

**Figure 6 molecules-27-06040-f006:**
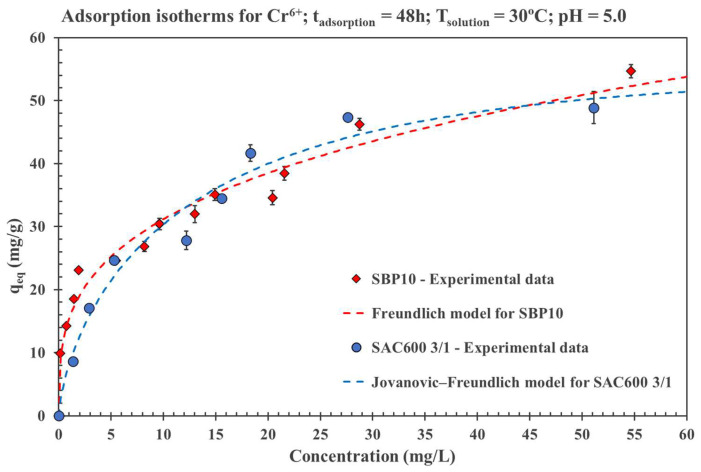
Adsorption isotherms for SAC600 3/1 and SBP10.

**Figure 7 molecules-27-06040-f007:**
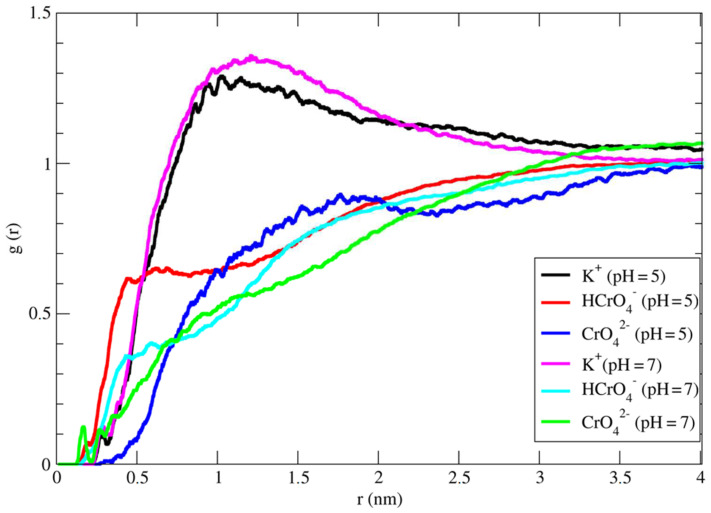
The radial distribution function of K^+^, HCrO_4_^−^ and CrO_4_^2−^ ions around activated carbon molecule.

**Figure 8 molecules-27-06040-f008:**
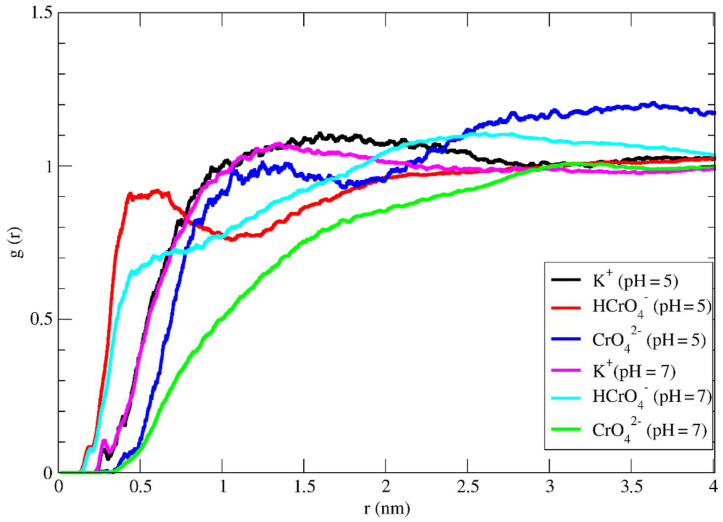
The radial distribution function of K^+^, HCrO_4_^−^ and CrO_4_^2−^ ions around activated carbon surface in the solution of MgCl_2_.

**Table 1 molecules-27-06040-t001:** Textural properties for SAC600 3/1 and SBP10.

Activated Carbons	S_BET_ (m^2^/g)	V_micro_ (cm^3^/g)	V_meso_ (cm^3^/g)	V_macro_ (cm^3^/g)	V_Tot_ (cm^3^/g)	D_p_ (nm)
SAC600 3/1	1695	0.727	0.773	0.339	1.838	4.34
SBP10	1900	0.823	0.502	0.013	1.338	2.82

**Table 2 molecules-27-06040-t002:** Total acidity and total basicity for lab-scale and pilot-scale Sargassum activated carbons.

Boehm Titration: Total Acidity and Total Basicity		
Sample Name	Acidic Groups (mg_eq_ H^+^/g)	Basic Groups (mg_eq_ OH^−^/g)
SAC600 3/1	103.62	20.40
SBP10	53.73	14.40

**Table 3 molecules-27-06040-t003:** Identification of absorption peaks of FTIR spectra for both AC samples.

Structure	Range (cm^−1^)	Peak (cm^−1^)	Probability	Assignment	References
RCH_2_CH_3_ (aliphatic sym.)	3000–2850	2980	Very possible	C-H stretching	[43,62]
RCH_2_CH_3_ (aliphatic asym.)	3000–2850	2903	Very possible	C-H stretching	[43,62]
Aromatic ring	1571–1561	1569	Very possible	C=C stretching	[43,62]
RCO-OH (carboxylic acids)	1320–1000	1184; 1081; 1066	Very possible	C-O stretching	[43,63]
RCOOR’ (esters)	1320–1000	1184; 1081; 1066	Possible	C-O stretching	[43,63]
Ar-OH (phenol)	1205–1195	1184	Very possible	C-O stretching	[43,62,63]
P=O (phosphine oxide)	1200–1100	1184	Possible	P=O vibration	[43,62,63]
P=O (phosphate)	1200–1100	1184	Very possible	P=O vibration	[43,62,63]
P=O (phosphate)	1200–1100	1184	Very possible	P=O vibration	[43,62,63]
C-O stretching	1081	1081	Very possible	C-O stretching	[55]

**Table 4 molecules-27-06040-t004:** Atomic concentration (%) and position BE (eV) of the chemical elements and functional groups present on the surface of the SAC600 3/1 and SBP10.

	SAC600 3/1		SBP10	
**Elementary Composition of the Surface**	**% Atomic Concentration**	**Position BE (eV)**	**% Atomic** **Concentration**	**Position BE (eV)**
C 1s	83.3	284.655	92.99	284.03
O 1s	13.14	533.105	6.52	531.98
N 1s	3.23	400.155	0.49	133.18
S 2p	0.18	163.855		
P 2p	0.15	133.655		
**Functional Groups from C 1s**	**% Atomic Concentration**	**Position BE (eV)**	**% Atomic** **Concentration**	**Position BE (eV)**
C=C, C-C, C-H	67.23	284.596	80.88	284.400
C-O	21.09	285.739	7.45	286.250
C=O	7.43	286.904	6.19	287.950
O-C=O	4.24	288.381	5.48	289.990

**Table 5 molecules-27-06040-t005:** Thermodynamic parameters.

Activated Carbon	T (K)	ln(K_eq_)	ΔG° (KJ·mol^−1^)	ΔH° (KJ·mol^−1^)	ΔS° (J·mol^−1^·K^−1^)
SAC600 3/1	298.15	0.6878	−1.70	−21.85	−67.3
	308.15	0.5379	−1.38		
	313.15	0.2292	−0.60		

**Table 6 molecules-27-06040-t006:** Adsorption kinetics parameters for SAC600 3/1 and SBP10.

Model	Parameter	Activated Carbon	
		SAC600 3/1	SBP10
Pseudo First-order Kinetics	*q_e_* (mg·g^−1^)	17.91	5.23
	*K_1_* (min^–1^)	0.0017	0.0030
	R^2^	0.8911	0.7209
Type 1—Pseudo Second-order Kinetics	*q_e_* (mg·g^−1^)	46.97	36.26
	*K_2_* (g/mg·min)	0.0004	0.0026
	R^2^	0.9973	0.9999
Type 2—Pseudo Second-order Kinetics	*q_e_* (mg·g^−1^)	39.10	34.84
	*K_2_* (g·mg^−1^·min^−1^)	0.0044	0.0138
	R^2^	0.6014	0.8307
Intraparticle Diffusion	*C_i_* (mg·g^−1^)	25.92	25.36
	*K_ip_* (mg·g^−1^·min^−0.5^)	0.4830	0.3844
	R^2^	0.5007	0.2463

**Table 7 molecules-27-06040-t007:** Adsorption isotherm parameters for SAC600 3/1 and SBP10.

Model	Parameter	Activated Carbon	
		SAC600 3/1	SBP10
Langmuir	*q_s_* (mg·g^−1^)	57.17	45.87
	*K* (L·g^−1^)	0.1207	0.3015
	* R^2^ *	0.9676	0.8384
	* RSS *	75.49	435.66
Jovanovic	*q_s_* (mg·g^−1^)	48.05	42.92
	*K* (L·g^−1^)	0.1020	0.1547
	* R^2^ *	0.9516	0.7451
	* RSS *	112.73	687.09
Fowler	*q_s_* (mg·g^−1^)	57.17	45.87
	*K* (L·g^−1^)	0.1207	0.3015
	*χ*	0.00	0.00
	* R^2^ *	0.9676	0.8384
	* RSS *	75.49	435.66
Freundlich	*K* (L·g^−1^)	12.0248	15.4435
	* ν *	0.3809	0.3048
	* R^2^ *	0.9504	0.9749
	* RSS *	115.51	67.56
Jovanovic-Freundlich	*q_s_* (mg·g^−1^)	54.54	58.94
	*K* (L·g^−1^)	0.0744	0.0601
	* ν *	0.7014	0.4363
	* R^2^ *	0.9728	0.9459
	* RSS *	63.28	145.94
Fowler-Guggenheim/ Langmuir-Freundlich	*q_s_* (mg·g^−1^)	64.27	37.80
	*K* (L·g^−1^)	0.1552	1.6365
	*χ*	0.3509	0.5405
	* ν *	1.00	1.00
	* R^2^ *	0.9719	0.9627
	* RSS *	65.53	100.61

## Data Availability

The data presented in this study are available in the manuscript and Appendix A.

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
