# Peer review of "Adsorption of Hexavalent Chromium Using Activated Carbon Produced from Sargassum ssp.: Comparison between Lab Experiments and Molecular Dynamics Simulations"

_molecules, 2022, doi:10.3390/molecules27186040_

Round 1
Reviewer 1 Report
molecules-1893051 – Adsorption of hexavalent chromium using activated carbon produced from Sargassum (sp): Comparison between lab experiments and molecular dynamic simulations.
1- The introduction could be improved by referring to the following refs including
https://doi.org/10.1021/acs.jced.0c00085
https://doi.org/10.3390/nano11081907
https://doi.org/10.3390/fib10010007
https://doi.org/10.3390/polym13193427
https://doi.org/10.1021/acsomega.0c03652
which can offer help to compose a better writing for the introduction and retrieve useful information regarding this work as well.
2- Analyses of data/results and focus of aims should be further elevated.
3- Try other adsorption isotherm models.
I recommend this work for publication after major revision.
Author Response
Manuscript ID: molecules-1893051
Response to Reviewers (06/09/2022)
Dear Editor,
We would like to thank you for your and reviewers comments and questions. We appreciate the time and effort you have dedicated to provide interesting feedback on our manuscript. We are grateful for the insightful comments and valuable improvements to our paper.
We have incorporated most of the suggestions made by the reviewers. The changes were tracked and are highlighted within the manuscript. Please see the modified manuscript.
Below, you will find the responses to the questions and comments of each reviewer.
Best regards,
Prof. Sarra Gaspard
- REVIEWER 1:
- The introduction could be improved by referring to the following refs including:
https://doi.org/10.1021/acs.jced.0c00085
https://doi.org/10.3390/nano11081907
https://doi.org/10.3390/fib10010007
https://doi.org/10.3390/polym13193427
https://doi.org/10.1021/acsomega.0c03652
which can offer help to compose a better writing for the introduction and retrieve useful information regarding this work as well.
Author response: The introduction was improved following the reviewer’s comments. Among recommended references only two of them were included:
https://doi.org/10.3390/nano11081907
https://doi.org/10.1021/acsomega.0c03652
The other three references proposed by the referee were not included because we consider that they are not relevant to our manuscript. However, to enrich the introduction, other three references were added considering their relevance to the studied subject:
https://doi.org/10.3390/app10248812
https://doi.org/10.3390/molecules25215156
https://doi.org/10.3390/chemistry2010002
Reviewer 2 Report
I have completed the evaluation of the paper “Adsorption of hexavalent chromium using activated carbon produced from Sargassum (sp): Comparison between lab exper-iments and molecular dynamic simulations” by Yeray Alvarez-Galvan et al. submitted to Molecules for its possible publication. The authors propose strategy for adsorption of hexavalent chromium by the transformation of Sargassum, into activated carbon (AC), and investigate the effect of different experimental conditions on the adsorption performance, and further clarify the mechanism by MD simulation. It can be published after addressing the following comments:
(1) “Molecular dynamic simulations” section: The authors should provide the information of program or software for producing MD simulations, and the detailed force field parameters of all molecules simulated should be given.
(2) The sections “Abstract”, “Introduction” and “Conclusions” should be carefully re-written because of unclear statements. The title should delete the abbreviation.
(3) Figures 7-8 should be re-written because the lines are sheltered. Figure 5 should be also re-written because the messy typography.
(4) All of the tables in the main text should be re-written in three-line style.
(5) In the main text, it should be ΔG, ΔS… rather than AG, AH…, it should be oC rather than oC.
(6) Please check the sentence “Error! Reference source not found” in the main text.
(7) All of the equations should be used in Italics.
(8) All of the references should be carefully checked because of the inconsistent style.
For example, some references lack the number of DOI.
3. Liranzo-Gómez, R.E.; García-Cortés, D.; Jáuregui-Haza, U. Adaptation and Sustainable Management of Massive Influx of Sar-gassum in the Caribbean. Procedia Env. Sci Eng Manag 2021, 8, 543–553.
23. Rambabu, K.; Banat, F.; Nirmala, G.S.; Velu, S.; Monash, P.; Arthanareeswaran, G. Activated Carbon from Date Seeds for Chromium Removal in Aqueous Solution. Desalin Water Treat 2019, 156, 267–277.
Author Response
Manuscript ID: molecules-1893051
Response to Reviewers (06/09/2022)
Dear Editor,
We would like to thank you for your and reviewers comments and questions. We appreciate the time and effort you have dedicated to provide interesting feedback on our manuscript. We are grateful for the insightful comments and valuable improvements to our paper.
We have incorporated most of the suggestions made by the reviewers. The changes were tracked and are highlighted within the manuscript. Please see the modified manuscript.
Below, you will find the responses to the questions and comments of each reviewer.
Best regards,
Prof. Sarra Gaspard
- REVIEWER 2:
- “Molecular dynamic simulations” section: The authors should provide the information of program or software for producing MD simulations, and the detailed force field parameters of all molecules simulated should be given.
Author response: We thank the reviewer for the comments made on the force field parameters. We have included information about the force field parameters of chromium anions, activated carbon, and ions in the manuscript.
The force field parameters for chromium ions and activated carbon molecule were described by the potential function based on General Amber Force Field (GAFF) parameters [Re-fA, RefB] while for Na+, Cl- and Mg2+ ions the ff99SB [RefC] amber force field which compatible with GAFF have been applied.
In addition, we added more detailed information on the force field parameters such as Lennard-Jones-potential, Bond parameters, valence-angle parameters, atom types, and atomic point charges in the supplementary materials.
[RefA]: Wang, J.; Wang, W.; Kollman, P.A.; Case, D.A. Automatic Atom Type and Bond Type Perception in Molecular Mechanical Calculations. J. Mol. Graph. Model. 2006, 25, 247–260, doi:10.1016/j.jmgm.2005.12.005.
[RefB] : Wang, J.; Wolf, R.M.; Caldwell, J.W.; Kollman, P.A.; Case, D.A. Development and Testing of a General Amber Force Field. J. Comput. Chem. 2004, 25, 1157–1174, doi:10.1002/jcc.20035.
[RefC]; James A. Maier, Carmenza Martinez, Koushik Kasavajhala, Lauren Wick-strom, Kevin E. Hauser, and Carlos Simmerling; ff14SB: Improving the Accuracy of Protein Side Chain and Backbone Parameters from ff99SB; Journal of Chemical Theory and Computation, 2015 11 (8), 3696-3713 DOI: 10.1021/acs.jctc.5b00255
Concerning the performing of the MD simulations, we have to mention that the Gromacs 4.6.5 program package was used for conducting all MD simulations where, in the reference [52,53], the program package is described in more detail.
- The sections “Abstract”, “Introduction” and “Conclusions” should be carefully re-written because of unclear statements. The title should delete the abbreviation.
Author response: Done. The different mentioned sections were carefully rewritten. The title has been corrected. Spp. is a recognised international nomenclature and refers to more than one unnamed species. The name was changed to italic as it is a Latin name. See the modified manuscript.
- Figures 7-8 should be re-written because the lines are sheltered. Figure 5 should be also re-written because the messy typography.
Author response: Done.
- All of the tables in the main text should be re-written in three-line style.
Author response: Done.
- In the main text, it should be ΔG, ΔS… rather than AG, AH…, it should be oC rather than o
Author response: Done.
- Please check the sentence “Error! Reference source not found” in the main text.
Author response: Done. This error appeared after using Zotero without the right reference. The problem has been solved.
- All of the equations should be used in Italics.
Author response: Done.
- All of the references should be carefully checked because of the inconsistent style. For example, some references lack the number of DOI.
Liranzo-Gómez, R.E.; García-Cortés, D.; Jáuregui-Haza, U. Adaptation and Sustainable Management of Massive Influx of Sar-gassum in the Caribbean. Procedia Env. Sci Eng Manag 2021, 8, 543–553.
Rambabu, K.; Banat, F.; Nirmala, G.S.; Velu, S.; Monash, P.; Arthanareeswaran, G. Activated Carbon from Date Seeds for Chromium Removal in Aqueous Solution. Desalin Water Treat 2019, 156, 267–277.
Author response: Done. The references have been carefully checked and the inconsistencies in the style were corrected. The DOIs were included for all papers that have them.
Reviewer 3 Report
This study is focused on a biomass-derived activated carbon adsorbent synthesized for efficient Cr 6+ removal. this study also deals with some parametric analysis by means of experiments and simulation. Although this article is interesting, it requires both major and minor modifications before it can be published.
Minor comments:
1. Mention the yield of synthesized activated carbon.
2. Some of the figures and tables calling in the text is not in the correct format. in the PDF, the "Error! Reference source not found" text has been displayed several times.
3. Figure 1. although the caption says there are (a) and (b); however, in the figure, no such alphabetic numbering can be found.
4. Why only Mg2+ ion's effect on Cr6+ adsorption was tested?
5. The last line in the FTIR analysis requires a proper citation.
6. In table 5 the delta symbols (Δ) are not correct.
7. Figure 7. and Figure 8. The Axis number format should be 0.0 instead of 0,0.
8. There is no mention of the regeneration of the samples. Is the activated carbon reusable?
Major comments:
1. The quality and adsorption capacity of the activated carbon largely depends on the elemental composition of the raw biomass and synthesized activated carbon. Therefore elemental analysis of the raw biomass and synthesized activated carbon are recommended.
2. For different experiments, different initial concentration of Chromium solution was taken. Can the authors justify the reason behind this?
3. The pore size distribution for both SAC600 3/1 and SBP10 seems similar in Figure 3. However, their corresponding N2 adsorption/desorption isotherms hysteresis is very much different from one another.
4. The authors concluded that the higher functionality might have caused the higher Cr6+ uptake in SAC600 3/1; however, this statement seems to be contradictory as kinetics in SAC600 3/1 is at least 1 order slower than the SBP10. Apart from that, the pore volume in SAC600 3/1 is significantly higher which can definitely increase the Cr6+ uptake at the equilibrium.
5. The manuscript contains 111 references which according to me redundant. Instead of citing multiple articles for one single piece of information, the authors can cite one relevant article. apart from that, the authors are also encouraged to cite relevant recently published articles discussing biomass-derived activated carbons such as doi.org/10.1016/j.ijheatmasstransfer.2019.118579,doi.org/10.1016/j.ceja.2021.100086, doi.org/10.1016/j.applthermaleng.2020.115361, doi.org/10.1016/j.ijheatmasstransfer.2014.10.012.
Author Response
Manuscript ID: molecules-1893051
Response to Reviewers (06/09/2022)
Dear Editor,
We would like to thank you for your and reviewers comments and questions. We appreciate the time and effort you have dedicated to provide interesting feedback on our manuscript. We are grateful for the insightful comments and valuable improvements to our paper.
We have incorporated most of the suggestions made by the reviewers. The changes were tracked and are highlighted within the manuscript. Please see the modified manuscript.
Below, you will find the responses to the questions and comments of each reviewer.
Best regards,
Prof. Sarra Gaspard
- REVIEWER 3:
- Minor comments:
- Mention the yield of synthesized activated carbon.
Author response: Done.
- Some of the figures and tables calling in the text is not in the correct format. in the PDF, the "Error! Reference source not found" text has been displayed several times.
Author response: Done.
- Figure 1. although the caption says there are (a) and (b); however, in the figure, no such alphabetic numbering can be found.
Author response: Done.
- Why only Mg2+ion's effect on Cr6+ adsorption was tested?
Author response: We thank the reviewer for this interesting comment on the effect of Mg2+ ions. It is correct that the addition of other ions such as Ca2+ from CaCl2 or Na+ ions from NaCl to the solution due to an increase in the ionic strength can influence the adsorption of Cr6+. We had the hypothesis that the addition of doubly charged ions not only influences the ionic strength of the solution but also improves the contact ion-pair formation of ions with carboxylate groups of activated carbon. Mg2+ ion, due to its higher charge density, can make stronger contact ion pairs, therefore, even in lower concentration can neutralize the negatively charged pores. As a result, more Cr6+ ions can be adsorbed by the activated carbon.
- The last line in the FTIR analysis requires a proper citation.
Author response: The last line of the FTIR analysis was removed.
- In table 5 the delta symbols (Δ) are not correct.
Author response: Done.
- Figure 7. and Figure 8. The Axis number format should be 0.0 instead of 0,0.
Author response: Done.
- There is no mention of the regeneration of the samples. Is the activated carbon reusable?
Author response: This article is the first part of a research project where all the basic studies on adsorption are performed. At this time, the scaling and applicability experiments are being conducted, which include the studies of the breakthrough curve, column adsorption, number of adsorption cycles (AC regeneration), and column washing method, as well as their verification with a real polluted wastewater system.
- Major comments:
- The quality and adsorption capacity of the activated carbon largely depends on the elemental composition of the raw biomass and synthesized activated carbon. Therefore, elemental analysis of the raw biomass and synthesized activated carbon are recommended.
Author response: The recommended analyses were not conducted as we do not have the equipment necessary to conduct them. It is true that the adsorption capacity depends on the elemental composition of the raw material and the synthesised activated carbon. However, these analyses are not mandatory as it can be observed in many recently published papers similar to this one. Please, see some references below:
https://doi-org.bu-services.univ-antilles.fr/10.1016/j.jcis.2020.06.031
https://doi-org.bu-services.univ-antilles.fr/10.1016/j.jece.2020.104795
https://doi-org.bu-services.univ-antilles.fr/10.1016/j.jece.2019.103273
https://doi-org.bu-services.univ-antilles.fr/10.1016/j.jece.2013.12.014
- For different experiments, different initial concentration of Chromium solution was taken. Can the authors justify the reason behind this?
Author response: Yes. During the first analyses, a mass of 100 mg of activated carbon was used for each sample. For those 100 mg of activated carbon, a 60 mg/L chromium solution was prepared. In order to reduce the consumption of raw materials (activated carbon and potassium dichromate) and the environmental impact caused by chromium residues, a smaller mass of activated carbon (10 mg) and a lower concentration of chromium solution ( 6 mg/L) were used for the rest of the experiments.
- The pore size distribution for both SAC600 3/1 and SBP10 seems similar in Figure 3. However, their corresponding N2 adsorption/desorption isotherms hysteresis is very much different from one another.
Author response: The graph was represented using cm3·g-1·nm-1 in the Y axis. For this reason, the differences in the relative volume cannot be appreciated. We have changed the units and now, Y axis represents the relative volume (cm3·g-1) and the differences between both activated carbons can be clearly observed. Thank you for the comment as it needed to be changed. We have also corrected the Table 1 where the volume of micropores was included and Dp values were corrected as a mistake was made while copying the data from the Excel file. See the modified manuscript.
- The authors concluded that the higher functionality might have caused the higher Cr6+ uptake in SAC600 3/1; however, this statement seems to be contradictory as kinetics in SAC600 3/1 is at least 1 order slower than the SBP10. Apart from that, the pore volume in SAC600 3/1 is significantly higher which can definitely increase the Cr6+ uptake at the equilibrium.
Author response: We thank you for this comment as it is true that the higher Cr6+ uptake showed by SAC600 3/1 is mainly caused by its pore volume rather than the higher number of functional groups. The comment was modified according to this.
- The manuscript contains 111 references which according to me redundant. Instead of citing multiple articles for one single piece of information, the authors can cite one relevant article. apart from that, the authors are also encouraged to cite relevant recently published articles discussing biomass-derived activated carbons such as
https://doi.org/10.1016/j.ijheatmasstransfer.2019.118579
https://doi.org/10.1016/j.ceja.2021.100086
https://doi.org/10.1016/j.applthermaleng.2020.115361
https://doi.org/10.1016/j.ijheatmasstransfer.2014.10.012
The redundant references were removed as suggested by the reviewer. We thank the reviewer for its recommendations concerning other references, however, none of the recommended papers is related to the subject of this study and for this reason, we did not include them. On the other hand, all suggested papers by the reviewer are from the same research group. We do not consider a good practice to use the role of referee for improving the number of citations of a specific author or research group.
Round 2
Reviewer 1 Report
After this revision it in the manuscript ID "molecules-1893051" this paper is accept in present form and can be a very good contribution to molecules.
Reviewer 3 Report
The authors have addressed the comments to my satisfaction.